# Colorectal Cancer: Disease Process, Current Treatment Options, and Future Perspectives

**DOI:** 10.3390/pharmaceutics15112620

**Published:** 2023-11-12

**Authors:** Amusa S. Adebayo, Kafilat Agbaje, Simeon K. Adesina, Oluwabukunmi Olajubutu

**Affiliations:** College of Pharmacy, Howard University, 2400 6th St NW, Washington, DC 20059, USA; kafilat.agbaje@bison.howard.edu (K.A.); simeon.adesina@howard.edu (S.K.A.); oluwabukunmi.olajub@bison.howard.edu (O.O.)

**Keywords:** colorectal cancers, progression, current treatment options, future prospects

## Abstract

Colorectal cancer (CRC) is one of the deadliest malignancies in the US, ranking fourth after lung, prostate, and breast cancers, respectively, in general populations. It continues to be a menace, and the incidence has been projected to more than double by 2035, especially in underdeveloped countries. This review seeks to provide some insights into the disease progression, currently available treatment options and their challenges, and future perspectives. Searches were conducted in the PubMed search engine in the university’s online library. The keywords were “Colorectal Cancer” AND “disease process” OR “disease mechanisms” OR “Current Treatment” OR “Prospects”. Selection criteria were original articles published primarily during the period of 2013 through 2023. Abstracts, books and documents, and reviews/systematic reviews were filtered out. Of over 490 thousand articles returned, only about 800 met preliminary selection criteria, 200 were reviewed in detail, but 191 met final selection criteria. Fifty-one other articles were used due to cross-referencing. Although recently considered a disease of lifestyle, CRC incidence appears to be rising in countries with low, low–medium, and medium social demographic indices. CRC can affect all parts of the colon and rectum but is more fatal with poor disease outcomes when it is right-sided. The disease progression usually takes between 7–10 years and can be asymptomatic, making early detection and diagnosis difficult. The CRC tumor microenvironment is made up of different types of cells interacting with each other to promote the growth and proliferation of the tumor cells. Significant advancement has been made in the treatment of colorectal cancer. Notable approaches include surgery, chemotherapy, radiation therapy, and cryotherapy. Chemotherapy, including 5-fluorouracil, irinotecan, oxaliplatin, and leucovorin, plays a significant role in the management of CRC that has been diagnosed at advanced stages. Two classes of monoclonal antibody therapies have been approved by the FDA for the treatment of colorectal cancer: the vascular endothelial growth factor (VEGF) inhibitor, e.g., bevacizumab (Avastin^®^), and the epidermal growth factor receptor (EGFR) inhibitor, e.g., cetuximab (Erbitux^®^) and panitumumab (Verbitix^®^). However, many significant problems are still being experienced with these treatments, mainly off-target effects, toxic side effects, and the associated therapeutic failures of small molecular drugs and the rapid loss of efficacy of mAb therapies. Other novel delivery strategies continue to be investigated, including ligand-based targeting of CRC cells.

## 1. Introduction

Globally, colorectal cancer (CRC) is the third most commonly diagnosed cancer (with approximately 1.9 million cases) and the second most common cause of death (with approximately 0.9 million deaths worldwide) in 2020 [1]. Thus, colorectal cancer (CRC) accounts for about 10% of cancer cases and deaths, thus constituting a major component of the global cancer burden [2]. Although CRC is presently considered primarily a disease of Western society, a more rapid increase in incidence is being recorded in countries experiencing economic development with the associated changes in lifestyle and diet [3]. In addition, while the availability of screening services and improved treatment is considerably reducing CRC incidence and mortality in affluent countries [4], available screening programs and existing medical services are grossly inadequate [5] for timely diagnosis and treatment of cases that could arrest the escalating increase in incidence and death rates in the less affluent countries. A study by Liu et al. [2] showed that, when visualized at the social demographic index (SDI) level, the trend of age-adjusted death rate (AADR) increased in the low, low–middle, and middle SDI regions while it decreased in the high SDI region.

Despite the existence of effective screening techniques in high SDI regions that could reduce mortality rates, CRC continues to be a significant health threat [6]. In the US, CRC is presently the fourth most diagnosed but still the second most deadly after lung cancer [7]. It is projected that in 2023, there will be 153,020 new cases of colon cancer and 52,550 new cases of rectal cancer in the United States and that the global incidence of CRC will more than double by 2035, with the greatest increase in CRC burden and death expected in less developed countries [7,8]. The diversity of CRC diagnosis, prognosis, and survival is well established among the US population. CRC mortality is 36% higher among African-American men than among non-Hispanic white men and 34% higher among African-American women than among non-Hispanic white women [9,10,11].

Histologically, adenocarcinoma is the most prevalent variant of colon cancer. Other histological types are scirrhous and neuroendocrine tumors [8]. Adenomatous polyposis coli (APC) is a CRC tumor suppressor gene found on the 5q21-q22 chromosome. APC is a colorectal cancer gene that is dysregulated at both the germline and somatic levels [12,13]. APC gene encodes a 310 kDa protein containing 2843 amino acids. About 75% of the coding sequence is located on exon 15, the most common region for both germline and somatic mutations of APC. APC’s germline mutations result in familial adenomatous polyposis (FAP), the major hereditary predisposition to CRC development [12,14]. On the other hand, somatic APC mutations are associated with more than 80% of sporadic colorectal tumors. Apart from CRC caused by CpG island methylator phenotype (CIMP) or mismatch repair hypermutable microsatellite instability (MSI) phenotype, alterations in the APC gene are the initiator of over 85% of sporadic colorectal cancers [13,14]. In addition, sporadic CRC cases, with no family history or inherited genomic alterations, comprise 60–65% of all CRC cases [15,16].

CRC arises from the epithelial cells lining the colon or rectum in the gastrointestinal tract due to abnormal growth of glandular epithelial cells [17]. The development of colorectal adenocarcinomas is caused by sequential genetic and epigenetic mutations in specific oncogenes in the epithelial cells of the gastrointestinal tract [18]. The normal epithelium forms into hyperproliferative mucosa and eventually forms a benign adenoma, which can then progress to carcinoma and metastasis over the course of about 10 years. This process occurs through various mechanisms, including microsatellite instability (MSI) and chromosomal instability (CIN) pathways [19].

In MSI, tumors are caused by the inactivation of one of the four mismatch repair (MMR)genes: MSH2, MLH1, MSH6, and PMS2. During normal DNA replication, these proficient MMR genes detect errors in DNA mismatches, and the MLH1 and/or PMS2 heterodimer helps remove these errors and form new, corrected DNA strands [20]. The human genome contains over 10,000 regions of short tandem repeat sequences called microsatellites, which are prone to replication errors and heavily rely on the MMR system for repair. Deficiency in the MMR system leads to a faster accumulation of genetic errors at these microsatellites, resulting in the formation of tumors [21]. In CIN, tumors are caused by mutations in genes that result in chromosomal aneuploidy, loss of heterozygosity, and structural chromosomal rearrangements. In addition to these karyotypic abnormalities, CIN tumors also have an accumulation of specific mutations in tumor suppressor genes and oncogenes that activate pathways crucial for the initiation and progression of CRC [22].

The most common biomarkers, which are overexpressed in the CRC cellular membrane, include the folate receptor (FR-α, 37.1%), epidermal growth factor receptor (EGFR, 32.8%), the carcinoembryonic antigen (CEA, 98.8%), and tumor-associated glycoprotein (TAG-72, 79%) [23]. A major biomarker that has been extensively reported to be overexpressed in CRC is the human transferrin receptor (hTfR). The human transferrin receptor (hTfR) is ubiquitous and overexpressed (>100 folds) in many cancers [24,25,26,27,28].

The most significant risk factor in the development of colorectal cancer is age, with adolescents and older individuals being more susceptible. Other risk factors include sedentary behaviors, Western-style diets, smoking, diabetes, obesity, and, most importantly, the occurrence of ulcerative colitis as a result of *H. pylori* infection (Figure 1) [29], which will be discussed later in this review. Although early-stage disease treatment yields the most favorable results, approximately half of the colorectal cancer patients are either diagnosed with metastatic disease, develop the advanced-stage disease later, or experience cancer recurrence within a few months [30]. For those with metastatic disease that cannot be treated through surgery, the prognosis is bleak, with a 5-year survival rate of roughly 14% [31,32]. The diagnosis of CRC is made using various methods and techniques with varying sensitivities (Figure 2). Studies have shown that individuals who undergo colonoscopy have a lower risk of developing CRC compared to those who do not get screened [33]. The effectiveness of sigmoidoscopy in reducing both the incidence and CRC-related death has been established, as well as the combination of sigmoidoscopy and stool testing, which are cost-effective approaches but, unfortunately, have not been well utilized in established guidelines [34,35].

## 2. Search Method

Searches were conducted electronically online, primarily in PubMed. The MESH words used for the search were (((((((((Colorectal Cancer) AND (disease process)) OR (disease mechanisms)) OR (Current Treatment)) OR (Prospects)) NOT (abstract)) NOT (books and documents)) NOT (review)) NOT (systematic review)). The result was filtered for the past 10 years (2013–2023). A total of 493,983 results were obtained. Article selection protocol is shown in Figure 3 below.

Additional searches were conducted in Google Scholar for any article that was found relevant due to the historical perspective of the study through citations in primary articles. Such articles have been included in order to avoid information gaps and to substantiate important claims by primary articles found in PubMed engine.

## 3. Results and Discussion

Figure 4 shows the distribution of published manuscripts on colorectal cancer based on the MESH terms used. An average of 54 thousand CRC-related manuscripts were published annually as retrieved by the MESH terms used. Annual number of published manuscripts on the subject of CRC rose steadily between 2013 and 2020. The highest number of annual publications was recorded in the year 2021.

### 3.1. The Colon and Peculiarities of Colorectal Cancer

The colon is an important part of the gastrointestinal tract and is located in the abdomen, just below the stomach. It is about 150 cm long; the starting point is called the caecum, and the ending point is the anal verge. The colon is divided into four segments, namely: caecum and appendix vermiformis, colon segments which are divided into ascending, transverse descending colon, sigmoid colon, and anal canal. The first two segments are referred to as the left colon, and the last two are called the right colon. These segments have varying diameters; the caecum is the widest (7.5 cm), and the sigmoid colon is the narrowest (2.5 cm) [36]. In terms of embryology, the colon arises from the midgut and hindgut; the midgut gives rise to the transverse colon, ascending colon, and caecum, while the hindgut gives rise to descending colon, sigmoid colon, and anal canal. A description is shown in Figure 5.

The colon’s primary function is to extract water, nutrients, and electrolytes from partially digested food and to further process the remaining material to form stool. The stool is stored in the rectum and eventually expelled from the body through the anus [37].

### 3.2. CRC Development Processes

CRC can be categorized based on the location of the tumor within the colon and rectum. Right-sided colon cancers are found in the transverse colon, ascending colon, and caecum, while left-sided colon carcinomas are found in the transverse colon, descending colon, sigmoid colon, and anal canal [38,39]. Based on this categorization, the majority (about 63%) of diagnosed CRC is left-sided and mostly in younger patients [40,41].

Disease progression of colorectal cancer is shown in Figure 6. The symptoms and the prognosis of CRC have been shown by several studies to be associated with the location of the tumor [42,43]. Symptoms associated with left-side colorectal cancer include partial or total intestinal obstruction due to the narrow lumen of the colon on this side. As a result of the obstruction, symptoms such as constipation, nausea, abdominal distention, and abdominal pain are common. Right-sided CRC usually results in gross rectal bleeding, and the patient may present with iron-deficiency anemia [44]. Patients with right-sided CRC usually have a poor prognosis and are prone to metastasis compared to left-sided tumors [45,46]. The need for early diagnosis and search for more effective therapies is underscored by the fact that the majority of CRC (approx. 50%) are diagnosed at an advanced stage with resulting poor prognosis and high mortality [47].

The colorectal tumor microenvironment (TME) is characterized by multiple distinct cell types, primarily originating from the adjacent mesenchymal stroma and their interaction with tumor cells. The interactions between the tumor cells and neighboring stroma cells contribute to the growth and invasion of the tumor [48]. Tumor cells release various signals that transform the surrounding microenvironment into a pathological entity, which undergoes continuous changes throughout the progression of cancer. This transformed TME is believed to influence abnormal tissue functions and significantly contribute to the development of resistance to treatment [49]. The TME is made up of Cancer Associated Fibroblast (CAF), which represents the highest number of cells in the TME, immune cells such as myeloid cells, mast cells, dendritic cells, and tumor-associated macrophages (TAM) [50]. CAF produces and releases factors such as hepatocyte growth factor (HGF), epidermal growth factor (EGF), insulin-like growth factors 1 and 2 (IGF1 and IGF2), fibroblast growth factor 2 and 7 (FGF2 and FGF7), and vascular endothelial growth factor (VEGF). These factors have significant effects on processes such as inflammation, invasion, immune evasion, and angiogenesis, all of which contribute to the promotion of tumor growth [51,52]. CAF is associated with poor prognosis and is an important marker for predicting remission in patients [53].

### 3.3. H. pylori and Colorectal Cancer

Discovered in 1982, *Helicobacter pylori* (*H. pylori*) has been characterized as a spiral-shaped microaerophilic Gram-negative bacillus [54,55]. Its global presence and putative modern and ancient migrations were mapped by Suerbauma and Achtmanb (2004) [56]. The spiral shape helps the cell traverse the sticky mucus lining to eventually lodge in the stomach muscle (Figure 7a), where it produces ***exotoxins*** that weaken the protective mucus layer of the stomach wall. Once established, *H. pylori* increases meal-stimulated gastrin levels, reduces gastric mucus production, and decreases duodenal mucosal bicarbonate secretion. All these mechanisms support its survival in the gastrointestinal tract, where it can live insidiously for many years.

Most carriers of *H. pylori* have been known to contract the organism in childhood, and it can last a lifetime if left untreated [55]. Its presence is often asymptomatic in the majority of people, and its symptoms are often masked by other git conditions. Major advances have been made in the diagnosis of *H. pylori* (Figure 8). Early manifestation of *H. pylori* pathogenicity includes acute or chronic gastritis or peptic ulceration. Zhao et al. [57] and Zuo et al. [58] reported an association between *H. pylori* infection and colorectal cancer from meta-analysis of the available literature and systematic reviews. The study by Zuo et al. [58] involved a systematic review and meta-analysis of the literature from forty-seven studies, comprising 12,146 cases of colorectal cancer and 55,811 control cases in populations from Asia, America, and Europe. It was concluded that *H. pylori* infection is modestly associated with a heightened risk of colorectal cancer.

Until very recently, the relationship between *H. pylori* infection and colorectal cancer continued to be the subject of conflicting reports [59,60,61] with divergent observations in different patient populations. Prevailing evidence strongly indicates that *H. pylori* infection poses a significant risk factor for the onset of colorectal cancer. Key virulence factors that heighten cancer development have been identified, including cytotoxin-associated gene A (CagA), sialic acid-binding adhesin A (SabA), blood group antigen-binding adhesin A (BabA), and vacuolating cytotoxin A (VacA) (Figure 7b) [55]. More recently, *H. pylori* has been recognized as a causative agent of gastric adenocarcinoma, mucosa-associated lymphoid tissue (MALT) lymphoma, and gastric non-Hodgkin’s lymphoma. In 1994, *H. pylori* was categorized by the World Health Organization (WHO) as a Class I Carcinogen [62,63], and in 2017, it was listed as a high priority drug-resistant bacterium [64,65].

**Figure 8 pharmaceutics-15-02620-f008:**
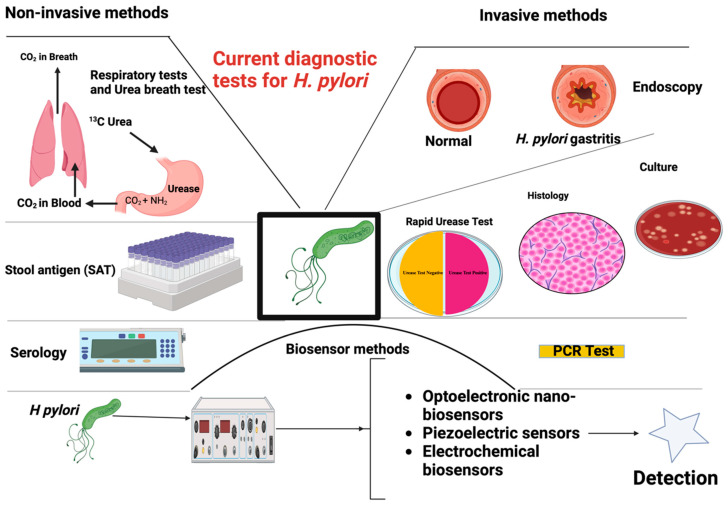
Diagnostic tests for *H. pylori* infection. Created with BioRender.com (A modification of original illustration by Cardos et al. [66]).

## 4. Treatment of Colorectal Cancer

The treatment options for CRC include elimination of underlying infections, surgery, cryosurgery, chemotherapy, radiation therapy, and targeted therapy [67], with surgery being the main form of curative treatment and often supplemented with chemotherapy (Figure 9).

### 4.1. Surgery

Surgery is the major treatment for colorectal cancer, especially where liver metastasis has occurred, and can be carried out using either the open or laparoscopic surgery technique [68]. Before proceeding with surgery, it is important to accurately determine the characteristics and extent of the tumor. Whenever feasible, it is recommended to obtain histological confirmation of the diagnosis, which may involve performing a colonoscopy if needed. This allows for a more precise understanding of the nature of the tumor and the stage of the disease, guiding the subsequent surgical approach [69]. The surgery involves the removal of the tumor and its surrounding mesentery, followed by the establishment of a primary connection between the remaining healthy sections.

The specific scope of the surgical removal is determined by factors such as the tumor’s position and the blood supply it receives [70]. As promising as surgery looks in the treatment of colorectal cancer, there are a lot of complications that arise post-surgery. These complications include but are not limited to infections, thrombosis, adhesion and small bowel obstruction, ileus, anastomotic leakage, and colonic ischemia. Some of these complications arise because of the complexities involved in the tumor removal process, the length of time it takes to conclude the removal, and the presence of other comorbidities [71].

In recent times, there has been an increase in the utilization of robotic surgery, which is less invasive. Robotic procedures provide distinct benefits compared to laparoscopic surgery. They enable an immersive three-dimensional (3D) visualization that enhances depth perception, thanks to the utilization of advanced imaging technology. The robotic system incorporates articulating wrists, allowing for enhanced dexterity and precision during surgical maneuvers. Moreover, the system eliminates the natural tremor that a surgeon may have, further enhancing accuracy. These advantages enable surgeons to perform more intricate and precise movements, enabling the completion of complex procedures [72].

### 4.2. Cryotherapy

Cryotherapy emerged as an adjunctive therapy for low rectal cancer during the 1970s. This therapeutic approach relies on the Joule–Thomson effect, which involves the ablation of tumors using low temperatures. Cryotherapy works by subjecting the tumor to extreme cold, leading to the destruction of cancer cells. It is considered a technique for tumor ablation based on the application of controlled low temperatures. Cryotherapy has been extensively utilized in the treatment of various tumors, particularly in the prostate, kidney, liver, and other solid organs [73]. However, there is limited information available regarding the application of cryoablation in patients with rectal cancers. In recent times, cryoablation has been considered a significant approach for treating patients with rectal cancers. This treatment method offers several advantages, including the preservation of the anus and protection of anal function. Additionally, cryoablation is considered a minimally invasive surgical procedure with low anesthesia requirements [74]. Several factors, such as the tumor site, tumor size, tumor stage, and lymph node metastasis, can influence the effectiveness of cryotherapy [75].

### 4.3. Radiation Therapy

The tumor stroma plays a crucial role in supporting tumor growth through various mechanisms, such as immunosuppressive cytokine secretion and metabolic alterations. It provides a complex network of cellular activities, such as preventing the physical contact of T cells and cancer cells, thereby facilitating tumor proliferation [76]. However, recent advancements in localized treatment, particularly radiation therapy, have enabled strategies to effectively target and eliminate tumor stromal tissue. Radiation therapy has the capability to induce immunogenic cell death, leading to the release of tumor antigens and promoting immune cell trafficking. This provides radiation therapy a unique advantage in counteracting the immune evasion mechanisms orchestrated by stromal cells within the tumor microenvironment [77].

While radiation therapy is not typically recommended as a routine treatment for colon cancer patients, it may have a role in specific situations. Radiation therapy may be considered for patients with locally advanced diseases that cannot be completely removed by surgery or are unlikely to achieve complete removal. Additionally, in cases of local recurrence, radiation therapy may also be utilized as a treatment option. It is important to note that the decision to use radiation therapy in these scenarios would be made on a case-by-case basis, considering the individual patient’s condition and factors such as tumor location and extent [78].

Intraoperative radiation therapy (IORT) is a valuable technique for increasing the radiation dose in patients with locally advanced primary and recurrent rectal cancer. It is typically combined with preoperative external beam radiation therapy (EBRT) and the administration of 5-fluorouracil (5-FU) or capecitabine. For patients with locally advanced primary or recurrent disease that can be completely removed through surgery, the potential benefits of dose escalation using IORT should be evaluated in prospective studies. This approach holds promise for improving treatment outcomes in these specific patient populations [79].

### 4.4. Chemotherapy

Chemotherapy involves the use of various drugs or combinations of drugs to slow down the rate of growth of cancer cells. However, conventional chemotherapy for colorectal cancer often leads to off-target effects resulting in unwanted side effects such as neutropenia, anemia, hand–foot syndrome, diarrhea, gastrointestinal toxicity, mucositis, nausea, vomiting, fatigue, hematologic disorders, and liver toxicity [80]. These side effects and the development of drug resistance mean that treatment outcomes to date have not been entirely satisfactory [81].

Chemotherapeutic agents typically work by destroying rapidly dividing cells or altering their growth. They can be administered individually or in combination and may be delivered intravenously to reach cancer cells throughout the body or targeted to specific areas affected by cancer [82].

Different chemotherapeutic agents work via different mechanisms, such as interfering with cell division and reproduction, inhibiting the molecules and nutrients required for cancer cell growth, inducing cell death (apoptosis), blocking the formation of new blood vessels that supply a tumor, cutting off its nutrients and prevention of DNA or RNA formation by mimicking nucleotides, known as antimetabolites, which are most effective during the S phase of the cell cycle [83].

Chemotherapeutic agents used in the treatment of CRC include 5-fluorouracil, irinotecan, oxaliplatin, and leucovorin. 5-fluorouracil (5-FU) has been widely used as a first-line treatment for cancer for many years, and it is thought to be particularly effective against colorectal cancer [84]. However, it has several significant drawbacks, including a short half-life, poor absorption due to the enzyme dihydropyrimidine dehydrogenase, and nonselective action against healthy cells in the gastrointestinal tract and bone marrow [85]. Researchers have developed a methacrylic-based copolymer-coated version of 5-FU to overcome its limitations by exploiting its pH-sensitive properties and hydrogel matrix [86]. Eudragit^®^ P-4135F and Eudragit^®^ RS100 are used to create microspheres for 5-FU [87].

Irinotecan (IRT) is a potent chemotherapy drug that targets an enzyme called Topo 1, which is present in many types of cells [88]. The drug works by killing rapidly dividing cells, including cancer cells, but it also affects non-tumor cells, such as blood cells, epithelial cells, and commensal bacteria [89]. As a result, treatment with IRT often causes side effects such as neutropenia, diarrhea, and gastrointestinal toxicities such as mucositis and liver injury, particularly non-alcoholic steatohepatitis [90]. These side effects can vary greatly among individuals depending on the severity of their condition [91].

In some published phase II clinical trials, irinotecan has been found to be effective in treating advanced colorectal cancer that did not respond to treatment with 5-fluorouracil (5-FU). Response rates in these trials were between 13 and 27%. Both a weekly regimen for 4 weeks followed by a 2-week rest and a once-every-3-week schedule produced similar response and survival rates. A biweekly regimen produced similar response rates but with similar side effects, such as delayed diarrhea, neutropenia, nausea, and vomiting. High-dose loperamide can help with the most common side effect of diarrhea. Irinotecan has also been studied as a single agent in newly diagnosed CRC and has shown response rates between 19 and 32% and a median survival time of around 12 months, similar to 5-FU and leucovorin. Two phase III studies have shown that irinotecan treatment improves survival compared to 5-FU infusion and provides the best supportive care in patients with 5-FU-refractory CRC. Current studies are focusing on using irinotecan in combination with other chemotherapy drugs, in combination with radiation therapy, and as adjuvant therapy for locally advanced CRC [88,92]. A combination therapy containing 5-fluorouracil and leucovorin has also been used in the treatment of CRC.

Leucovorin has been shown to enhance the effectiveness of fluorouracil in laboratory studies by increasing the inhibition of an enzyme called thymidylate synthase. This was first reported in clinical studies by Machover in 1982 [93]. A meta-analysis of various studies also showed that a combination of fluorouracil and leucovorin improved both the response rate of tumors and overall survival in patients with metastatic colorectal cancer (mCRC) compared to using fluorouracil alone [94]. Ibrahim et al. [95], showed that the development and optimization of nanoparticles containing a combination of 5-fluorouracil (5-FU) and leucovorin (LV) led to improved interactions with colon and colorectal cancer cells in laboratory experiments. In the in vitro experiments, the colon-targeted nanosystem prevented drug release in the gastrointestinal regions and increased its release in the colorectal sites, thereby improving the anti-cancer activity of the drug combination. The authors proposed that this approach could help in lowering the administered dosages of the anti-cancer drugs and, at the same time, enhancing therapeutic efficacy while minimizing associated toxic side effects for patients.

### 4.5. Monoclonal Antibodies

The advanced knowledge of the molecular basis of cancer development has given rise to the discovery and development of drugs that target specific molecular biomarkers, such as monoclonal antibodies [96]. Monoclonal antibodies have gained wide popularity in the treatment of colorectal cancer. They are targeted agents that bind to ligands or the extracellular compartments of receptors. As a result of this binding, several signal transduction pathways necessary for the thriving of cancer cells are inhibited [97]. There are two classes of monoclonal antibodies approved by the FDA for the treatment of colorectal cancer; they are the vascular endothelial growth factor (VEGF) inhibitor, e.g., bevacizumab (Avastin^®^), and the epidermal growth factor receptor (EGFR) inhibitor, e.g., cetuximab (Erbitux^®^) and panitumumab (Verbitix^®^).

Bevacizumab is a humanized monoclonal antibody, and it acts by binding to VEGF, thus preventing it from binding to the VEGF receptor, thereby halting the signal transduction pathway that gives rise to angiogenesis, proliferation, and migration of cancer cells [98]. Cetuximab, which is a chimeric immunoglobulin antibody, and panitumumab, a fully human immunoglobulin antibody, both act by binding competitively to the extracellular domain of the EGFR, thus preventing the binding of EGF, which is known to play a significant role in angiogenesis and proliferation of cancer cells [99]. In addition to being an EGFR inhibitor, cetuximab has been shown to inhibit the production of pro-angiogenic factors such as VEGF. These groups of drugs have been evaluated in several clinical trials either as monotherapy, in combination with chemotherapy, or in combination with each other.

### 4.6. Side Effects and Complications of CRC Treatments

Classical chemotherapeutic agents for CRC treatment are fluoropyrimidines intravenous 5-fluorouracil (i.v. 5-FU) or oral capecitabine in various combinations and schedules. Constipation, diarrhea, dyspepsia, mucositis, nausea, and vomiting are the most common gastrointestinal (GIT) adverse events of these classical chemotherapeutic agents. Incidence and severity of adverse drug effects associated with these agents often vary with the drug and the therapeutic schedules, including dose, route of administration, frequency, co-administration of other agents, and overall duration of treatment [100]. GIT toxicities are the most concerning, and they tend to impact the patients’ compliance with therapeutic regimens. Infused regimens of 5-FU and leucovorin (LV) have shown less toxicity than bolus regimens and therefore preferred by patients. Previous efforts in the search for ideal 5-FU led to the discovery of floxuridine and tegafur, two typical 5-FU derivatives with enhanced pharmacokinetics and pharmacological profiles resulting in improved absorption, higher selectivity, and bioactivity, metabolic stability, and lower toxicity [101]. The emergence of multi-drug resistant cancer cells makes combination therapy, with agents that target different cancer progression pathways, the hallmark of cancer treatment. However, the potential for adverse drug events and toxicities increases with a number of components in a regimen. Most common combination therapies employ irinotecan and oxaliplatin with 5-fluorouracil and novel drugs targeting epidermal growth factor receptor (EGFR, such as cetuximab and panitumumab) or vascular endothelial growth factor (VEGF, such as bevacizumab, aflibercept, regorafenib, and ramucirumab) [100,102]

In a randomized clinical trial on primary CRC tumor resection plus chemotherapy versus chemotherapy (mFOLFOX6 plus bevacizumab or CapeOX plus bevacizumab) alone for patients with asymptomatic, synchronous unresectable metastasis by Kanemitsu et al. [103], adverse events for treatment discontinuation were based on the severity of the events, whether patient declined treatment, adverse effect delayed medication for more than 28 days, necessitated subsequent dose reduction after the second reduction, or if the patient declined treatment due to AEs [103]. Chemotherapy’s AEs may also be confounded by tumor-associated complications of intestinal obstruction, perforation, fistulation, or hemorrhage [103]. The classical PK difference across ages may have implications for dose titration or drug product selection. However, Kim [102] concluded that elderly patients with stage 3 colon cancer can be treated with adjuvant chemotherapy comprising 5-fluorouracil-leucovorin or 5-FU capecitabine without significant differences in toxicity from younger patients. Further, combination chemotherapy appears to be safe in older patients with metastatic CRC, except that for frail or vulnerable elderly patients, a stop-and-go strategy or monotherapy may be more appropriate.

## 5. Colon-Specific Drug Delivery Systems

Despite the availability of highly potent chemotherapeutic agents used to treat CRC, their low specificity and off-target cellular interaction often produce a range of dose-limiting side effects [23]. Thus, targeting CRC is essential. Most orally administered drugs are small molecules because the physiological barriers of the GIT do not support large molecular weight peptides and protein absorption [104,105]. Fortunately, the location of CRC posits it for passive targeting of cytotoxic agents via enteric coating to the colon. One major concern had always been physiologic, enzymatic, and GIT motility barriers to protein and peptide delivery via the enteron. The recent development of synthetic techniques for making small, modifiable, and stable peptide ligands for targeting drug molecules on the surface of CRC cells has created an incredible opportunity for colon-specific delivery of small, cytotoxic molecules that are conjugated to target ligands [26,106,107].

There is growing interest in the development of new drug delivery systems that can deliver drugs at a controlled rate and target specific sites of action. Conventional dosage forms, including extended-release forms, are limited in their ability to meet these goals [108]. Also, the diversity of cancer cells and the intricate mechanisms involved in tumor development have made current chemotherapy treatments for colorectal cancer ineffective. The current therapies usually involve high doses of toxic drugs, which cause severe side effects and make patients discontinue therapy [109]. Therefore, it is crucial to develop targeted drug delivery methods that are more effective in fighting tumors and less harmful to the patient [110]. Colon-specific drug delivery systems are promising and have shown to be effective for directing treatment to the site of disease, reducing the dosage, and minimizing systemic effects [111]. Formulations for colonic delivery are also ideal for administering drugs that are polar, easily degraded chemically or enzymatically in the upper gastrointestinal tract, or are heavily metabolized by the liver, such as therapeutic proteins and peptides [110].

However, progress in this area has been hindered by instability in physiologic fluids, low specificity of targeting, and off-target cellular interaction with associated dose-limiting side effects.

### 5.1. Approaches for Colon-Specific Drug Delivery

#### 5.1.1. PH Approach

The pH approach to CRC drug delivery is illustrated in Figure 10. This approach takes advantage of the pH gradient present in the gastrointestinal tract, which becomes more alkaline as it moves from the stomach (pH 1.5–3.5) and small intestine (5.5–6.8) to the colon (6.4–7.0) [112]. By using polymers that dissolve at different pH levels and understanding their properties, drug delivery systems can be created to target specific areas [113]. pH-sensitive polymers such as cellulose acetate phthalates (CAP), hydroxypropyl methyl-cellulose phthalate (HPMCP) 50 and 55, and copolymers of methacrylic acid and methyl methacrylate, such as Eudragit^®^ S 100, and Eudragit^®^ L, have been widely used [114]. The most used pH-sensitive polymers are the copolymers of methacrylic acid and methyl methacrylate, especially Eudragit, because they have the added advantage of being mucoadhesive [115,116]. The broad usage of polymers in drug delivery has been due to their exceptional properties that have yet to be matched by any other material [117].

Using pH-sensitive polymers of methyl methacrylic acid, known as Eudragit R and S100, as a targeting technique for the colon is an effective method. This is because these polymers have a dissolution starting point that is like the pH of the colon, and they also have a number of benefits, such as reducing the impact of pH and decreasing the risk of high-dose dumping. Additionally, these polymers can improve the uptake and bioavailability of acid-labile drugs that are difficult to dissolve. Eudragit S100 specifically is known to inhibit drug release in the terminal ileum and increase the selective release of drugs in the colon [118,119].

Prajakta et al. [120] designed a pH-sensitive curcumin-loaded Eudragit S100 polymer for drug delivery to an HT-29 colorectal cancer cell line and observed a 2-fold inhibition of the cancer cells when compared to the use of curcumin as a free drug. The authors attributed the improved activity to improved cellular uptake of the drug-loaded polymer by the cancer cells. As promising as the pH approach may appear, a major setback has been the intraindividual variation in the GIT pH values and the lower colonic pH levels in individuals suffering from ulcerative colitis and Chron’s Disease [121]. Aside from the use of polymers, hydrogels have also been investigated for use in pH-specific colon drug delivery. In a study, Liu et al. [122] synthesized and characterized a novel pH-sensitive semi-interpenetrating polymer network (semi-IPN) hydrogels based on starch and poly-aspartic acid (PAsp). The 5-FU-loaded hydrogels had excellent drug release properties attributed to the hydrogels’ pH-sensitive properties, which posits them for application in the drug delivery platform for colon cancer management [122]. Stimuli-response hydrogels, carbon nano-onions, and functionalized carbon nano-onions designed for pH-triggered drug delivery [123,124,125] also hold great potential for facilitating targeted delivery of drugs to CRC cells.

#### 5.1.2. Microbiota-Based Drug Delivery

The human body is inhabited by a diverse range of microorganisms, including bacteria, archaea, viruses, fungi, protozoa, and helminths, collectively known as the human microbiota. These microorganisms are increasingly recognized as significant factors in human health and disease causation [126]. The colon harbors more than 400 variety of microorganisms, including aerobic species such as *Escherichia coli* and anaerobic species such as *Clostridium* spp. These bacteria possess multiple metabolizing enzymes that are both hydrolytic and reductive in nature [112]. Microbiota-activated delivery systems have been identified as a potential solution for colon-targeted drug delivery. It is a well-explored approach based on the ability of specific enzymes of colonic bacteria to degrade xenobiotics, polysaccharides, and polymers (Figure 11). Colonic bacteria enzymes such as azoreductases, glucuronidase, glycosidases, esterases, and amidases are responsible for the breakdown of drug-loaded polymers and polysaccharides.

Polysaccharides, such as pectin, guar gum, inulin, and chitosan, are utilized due to their ability to remain intact in the upper gastrointestinal tract while being susceptible to degradation by colonic microflora, thereby releasing the loaded drug [119]. The properties of the polysaccharides, such as drug release rate, stability, and mucoadhesion, can be enhanced by structural modification of the polysaccharides [127]. Mucoadhesion is particularly important because it promotes drug uptake by ensuring prolonged contact between the drug delivery system and the mucosal surface [128]. The use of combinations of polysaccharides also offers more advantages in terms of drug delivery rate. Zhu et al. [129] designed a colon-targeted drug delivery system utilizing a combination of 2 polysaccharides, porous starch, pectin, and chitosan. The drug, doxorubicin, was first loaded on the porous starch, which was then coated with pectin and chitosan solution. The result showed a 4-fold increase in the drug release rate from the porous starch–pectin–chitosan–doxorubicin combination when compared to pectin-doxorubicin.

Advantages of these polysaccharides include the presence of numerous modifiable functional groups, large-scale availability, relatively low cost, low toxicity and immunogenicity, high biocompatibility, and biodegradability [130,131]. Despite the beneficial attributes of polysaccharides in microbiota-based delivery, some drawbacks have limited its use, including a broad range of molecular weight due to the varying chemistry of polysaccharides [132], low solubility in a wide range of organic solvents and excessive aqueous solubility which leads to early drug release in the upper GI tract [133]. To address these challenges, especially that of premature release in the upper GI tract, polysaccharides-based systems can be formulated by combining polysaccharides with other polymers. For instance, water-insoluble polymers such as Eudragit RS and ethyl cellulose can be used in combination with different polysaccharides to achieve colon-specific drug delivery.

Although the rate and extent of drug release in polysaccharide-based microbiota-triggered systems is appropriate in terms of colon specificity, factors such as changes in microbiota due to disease conditions, enzymatic secretions, gut infections, and dietary habits cannot ensure the precise degradation of microbiota-triggered delivery systems and, as such, may not produce consistent and reliable and predictable drug delivery pattern [134].

#### 5.1.3. Drug Conjugates

To treat colonic diseases with fewer toxic side effects, scientists are exploring drug conjugates that target specific receptors at disease sites through interaction between ligands on the carrier and expressed molecules on receptors [135]. Drug conjugates are compounds resulting from the chemical linkage between a drug and another molecule or compound.

This coupling aims to amplify the therapeutic impact, augment selectivity, or enhance the pharmacokinetic characteristics of the drug (Alas et al., 2020) [136]. Different ligands, such as antibodies, peptides, folic acid, and hyaluronic acid, can be chosen based on their interaction with certain molecules that are expressed on receptors (Figure 12). Table 1 highlights several receptors that have been targeted in the delivery of drug conjugates for the treatment of colorectal cancer. Candelaria et al. (2021) [137] discussed the potential for transferrin receptor 1 (TfR1) targeting as direct anticancer agents. Since rapidly dividing cancer cells require an abundance of iron intake, TfR1, a type II transmembrane glycoprotein, is usually overexpressed in cancer cells. Thus, designing antibodies that could directly disrupt the TfR1-dependent cellular supply of iron or antibody–drug (oncology) conjugate that can be internalized by TfR1 receptor-mediated endocytosis are promising mechanisms for CRC treatment. Drug conjugate systems can also be combined with pH-sensitive systems for improved stability and specificity in the gastrointestinal tract (Tesauro et al., 2019) [138].

Peptides have attracted significant attention as possible ligands for targeted drug delivery due to their biocompatibility, cost-effectiveness, chemical diversity, and responsiveness to stimuli (Ghosh et al., 2018, Jiang et al., 2019) [139,140]. Due to their large binding interfaces with receptors, peptide ligands also have higher binding affinity and specificity than small molecule ligands. They are easily accessible through high-throughput screening and can be synthesized using automated devices, and any observed instability can be addressed by modifying the peptide sequences (Al-azzawi et al., 2019, Ren et al., 2016) [141,142]. Peptide-conjugated drug delivery systems are particularly promising for tumor-targeted delivery. For example, the TK peptide (TWYKIAFQRNRK) has a high affinity for integrin, which is upregulated in colon cancer cells and was conjugated to doxorubicin-loaded micelles for colon-specific delivery. The TK-conjugated micelles showed improved cytotoxicity and penetration in tumor spheroids, making TK a promising targeting ligand for colon-targeted therapy (Guo et al., 2019) [143]. Table 2 Illustrates examples of experimental drug conjugates in clinical trial.

Top-of-form nanoparticles specifically designed for the colon were modified with both amphipathic chitosan derivatives (ACS) and cell-penetrating peptides (CPP) to enhance the oral effectiveness of insulin. The ACS modification protects the CPP from degradation in the upper gastrointestinal tract and ensures colon-specific drug delivery. Once the nanoparticles reach the colon, the ACS on their surface degrades, and the exposed CPP allows for the drug to penetrate the colonic epithelium. In vitro and in vivo evaluations indicate that these ACS-CPP nanoparticles may be a promising method for improving the oral absorption of proteins and peptides in a colon-specific manner (Langel, 2019) [144]. 

**Table 1 pharmaceutics-15-02620-t001:** Molecular targets and their ligands in colorectal cancer.

Targets	Ligands	References
Carcinoembryonic antigen	Antibody	Tiernan et al., 2013 [145]
Folate receptor alpha (FRα)	Folic acid	Noe et al., 2022 [146]
Human epidermal growth factor 2 (HER-2)	Peptide	Suwaidan et al., 2022 [147]
Hyaluronic acid receptor	Hyaluronic acid	Mansoori et al., 2020 [148]
CD44 receptor	Hyaluronic acid	Tabasi et al., 2021 [149]

**Table 2 pharmaceutics-15-02620-t002:** Drug conjugates in clinical trial.

Drug Conjugate	Target	Phase of Study	Clinical Trial Identifier	Sponsor
Disitamab vedotin and Tisleizumab	HER-2	II	NCT05493683	The First Affiliated Hospital with Nanjing Medical University (Nanjing, China)
M9140	Carcinoembryonic antigen	I	NCT05464030	EMD Serono Research & Development Institute, Inc. (Darmstadt, Germany)
CBP-1019	Unknown	I, II	NCT05830097	Coherent Biopharma (Hefei) Co., Ltd. (Hefei, China)
BDC-1001 (Nivolumab)	HER-2	I, II	NCT04278144	Bolt Biotherapeutics Inc. (Redwood City, CA, USA)
ELU001	(FRα)	I, II	NCT05001282	Elucida Oncology (Monmouth Junction, NJ, USA)
Datopotamab deruxtecan	Unknown	II	NCT05489211	AstraZeneca (Minato City, Janpan)
SBT6050 (Cemiplimab)	HER-2	I, IB	NCT04460456	Silverback Therapeutics (Seattle, WA, USA)
TORL-3-600	Unknown	I	NCT05948826	TORL Biotherapeutics, LLC (Albuquerque, NM, USA)

#### 5.1.4. Vesicular Systems

Vesicular systems are a popular method for targeted delivery and improved outcomes. They are nanocarriers and have gained wide popularity due to their benefits, such as extending the half-life of drugs and evading uptake by reticuloendothelial systems (RES) (Kadam et al., 2012) [150]. Lipid-based vesicles have a wide range of uses in fields such as immunology, membrane biology, diagnostics, and genetic engineering. They can be used to model biological membranes and to transport and target active pharmaceutical ingredients (API) (Barani et al., 2020, Bagheri et al., 2014) [151,152]. A list of vesicular systems for drug delivery to colorectal cancer currently in clinical trial is shown in Table 3.

##### Liposomes

Liposomes are colloidal, spherical structures that form through the self-assembly of amphiphilic lipid molecules in a solution, such as phospholipids (Sebaaly et al., 2016) [153]. Liposomes have a size range of 30 nm to the micrometer scale, and their phospholipid bilayer is typically 4–5 nm thick. The liposomal membrane can consist of one or more lipid bilayers arranged around an internal aqueous core. The polar head groups are oriented towards both the inner and outer aqueous phases. This organized structure provides liposomes with a unique ability to load and transport molecules with various solubilities, such as hydrophilic molecules within the internal aqueous core, hydrophobic molecules within the lipid bilayer, and amphiphilic molecules at the interface between the water and lipid bilayer (Laouini et al., 2012) [154].

Liposomes have been used as a drug delivery carrier in a variety of disease conditions, especially cancer. There are several FDA-approved liposomes-based anticancer drugs, such as Doxil^®^ (doxorubicin hydrochloride) used in the treatment of ovarian cancer and myeloid melanoma, Marqibo^®^ (Vincristine sulfate) used in the treatment of leukemia and Onivyde^®^ (irinotecan hydrochloride) used in the treatment of pancreatic adenocarcinoma (Liu et al., 2022) [155]. Liposomes are excellent drug carriers due to their remarkable properties, including the ability to shield the loaded drug from physiological degradation, prolong the drug’s half-life, and regulate of drug release rate (Niu et al., 2012, Wang et al., 2009) [156,157] and the ability to deliver chemotherapeutic agents to their site of action by active or passive targeting (Alavi, 2019) [158]. In colon-specific drug delivery, liposomes have been used as carriers to deliver the payload to tumor cells in the colon and to improve the efficacy and reduce side effects of existing chemotherapeutic agents, but most of the formulations are still in the experimental stage.

Khuntawee et al., 2021 [159] encapsulated cordycepin, a colon cancer drug candidate in liposomes, with the aim of overcoming the limitations of cordycepin, including poor stability and low water solubility. In vitro studies show that liposomal cordycepin inhibited the growth of the HT-29 colorectal cancer cell line, inducing apoptosis at a rate that was 2-fold greater than what was observed with free cordycepin. Xiong et al., 2017 [160] designed a mannosylated liposome loaded with paclitaxel for active targeting to mannose receptors, which are highly expressed in CT26 colon cancer cell lines. The result showed that there was increased cellular uptake of the mannosylated liposomes, and no toxicity or side effect was observed.

##### Niosomes

Nonionic surfactant vesicles (niosomes) are bilayer structures formed from the self-assembly of nonionic surfactants in an aqueous medium that requires external energy or heat to make the system stable (Yasamineh et al., 2022) [161]. The use was first reported in the cosmetic industries in the 1970s but has now gained wide popularity as an effective carrier for drugs to specific sites of action (Bagheri et al., 2014) [152]. Being bilayer vesicles, they are divided into unilamellar and multilamellar vesicles (Marianecci et al., 2014) [162]. Typically, niosomes fall within the sub-micron (colloidal) size range. Small unilamellar vesicles (SUVs) have particle sizes ranging from approximately 10 to 100 nm, while large unilamellar vesicles (LUVs) have sizes ranging from 100 to 3000 nm. Multi-lamellar vesicles (MLVs) are larger than 5 µm, and there have been a few reports of “giant” vesicles (>15 µm) as well (Figure 13) (Moghasemmi et al., 2014, Varshosaz et al., 2014) [163,164].

Niosomes are comparable to liposomes, but while liposomes contain phospholipids, niosomes are made from non-ionic surfactants like tweens and spans. Like other vesicular drug delivery systems, niosomes can entrap both hydrophilic and lipophilic drugs in either the aqueous layer or the lipid bilayer (Srinivas et al., 2010, Shakya et al., 2014) [165,166]. Niosomes are excellent drug carriers. Their advantage over other vesicular drug delivery systems includes ease of preparation, the ability to encapsulate a wide range of chemotherapeutic agents (Chen et al., 2019) [167], no need for covalent bonding of encapsulated payload, greater stability, and the ability to evade sequestration by the reticular endothelial systems and prolong circulation half-life (Witika et al., 2022) [168]. In terms of clearance, liposomes are more rapidly phagocytosed by cells of the RES and macrophages (El-Ridy et al., 2012) [169].

Several methods have been devised for niosomes preparation. Available methods of preparation include thin-film hydration (TFH), reverse-phase evaporation (RPE), evaporation/sonication (EVP/SON), and the ether injection method (EIM) (Umbarkar, 2021) [170]. The method used can have profound effects on the physicochemical properties of prepared niosomes. Ugorji et al., 2022 [171], prepared 5FU-loaded niosomes with either Tween 80 or Span 60 using the four methods mentioned above. The result showed that the TFH method produced the noisome with the smallest particle size (187 nm), while the EVP/SON produced the largest noisome with a particle size of 4476 nm. The particle size, polydispersity index, entrapment efficiency, and physical stability of the vesicles were influenced by the techniques used in their preparation. The thin-film hydration process produced smaller particle sizes and higher entrapment efficiency of 5FU than the other methods.

To evaluate the efficacy of 5FU-loaded noisome in colon cancer cell lines, Ugorji et al. (2022) [171] conducted a real-time cell assay, which is an in vitro cytotoxicity test against HCT-11 colon cancer cell lines. The result showed that compared to the pure drug and empty niosomes, the real-time cell assay indicated that the niosomes loaded with 5-fluorouracil induced more long-lasting cell death. Also, El-far et al., 2022 [172], designed oxaliplatin and paclitaxel-loaded noisome in a bid to assess the efficacy of the drug combination in HT-29 colorectal cancer cell lines. The result showed that as a distinctive nano micellar system, niosomes loaded with drugs can improve cellular absorption of drugs, leading to increased cytotoxicity and apoptosis of the cells. The drug-loaded noisome exhibited improved bioavailability and can be deemed as a promising alternative delivery system for these two anticancer drugs in the treatment of colorectal cancer.

##### Polymeric Nanoparticles

To surmount the associated challenges of poor specificities in most chemotherapeutic agents’ challenges, various substances such as polymeric nanoparticles and materials based on organometallic compounds and carbon have been employed as carrier systems for colorectal cancer treatment (Zhu et al., 2019; Rashmi et al., 2015) [129,173]. The emergence of nanotechnology has provided significant progress in the delivery of drugs to cancer cells. Although only a limited number of drugs in nanocarriers have made it through for clinical use, it is anticipated that nanocarriers will be a major delivery system to obtain the breakthrough formulation for the diagnosis and treatment of colorectal cancer in the foreseeable future.

Polymeric nanoparticles provide the opportunity to encapsulate drugs with varying properties, protect the drugs from degrading enzymes, and possess numerous other benefits compared to larger molecular-weight drugs. Moreover, modifying their surface with targeting molecules enhances their adeptness in navigating the in vivo surroundings and prolongs the controlled discharge of the payload. Considering that conventional nanoparticles (NPs) can easily be taken up by cells of the RES, strategies such as surface modification of the nanocarriers have been employed to slow down or prevent RES sequestration on NPs. This is achieved by coating the surface of the nanocarrier with a hydrophilic polymer such as polyethylene glycol (PEG), which protects the surface charge of the nanocarrier (Figure 14), thus making it unrecognizable to the RES (Lampretch, 2015) [174]. The optimal size of nanoparticles is typically within the range of 10 nm–200 nm; this size and their stealth property (due to coating with PEG) confers on them the ability to intravasate the leaky vasculature of cancer cells through the enhanced permeability and retention (EPR) effect (Greish, 2010; Ejigah et al., 2022) [175,176]. More recently, stimuli-responsive nanoparticles (nano-onions) comprising poly 4-hydroxyphenyl methacrylate–carbon nanoparticles embedding γ-cyclodextrin, gelatin methacrylate or bovine serum albumin nanocomposite fibers were designed by Mamidi et al. (2020, 2021a,b) [123,124,125]. Ionic gelation, primed layer-by-layer self-assembly, and Forcespinning^®^ technology were used as construction techniques. Controlled, pH-responsive release of model drugs (e.g., diclofenac, doxorubicin) was achieved.

Numerous polymers, including poly(lactide-co-glycolide) (PLGA), polylactide (PLA), polycaprolactone (PCL), poly(D,L-lactide), chitosan, and PLGA–polyethylene glycol (PEG), have been used to formulate nanocarriers to facilitate targeted delivery of chemotherapeutic agents to cancer cells. Among these, PLGA stands out as a highly suitable biodegradable polymer for drug delivery and biomaterial applications. This is attributed to its capability to undergo hydrolysis within the body, resulting in the formation of metabolite monomers—lactic acid and glycolic acid. As these monomers are naturally occurring and efficiently metabolized by the body, employing PLGA for medical purposes carries minimal risk of systemic toxicity (Prabhu et al., 2015) [173].

## 6. Prospects: Individualized Therapy and Drugs in Clinical Trials

The occurrence of colorectal cancer continues to increase, especially in young adults. The disease burden is higher in developing countries with no proper diagnostic techniques and equipment. Though there are currently available therapies, most of them come with various challenges, such as toxicity and severe complications. The outlook for effective treatment of colorectal cancer will be to individualize treatment, such that each patient is treated based on the peculiarities of their situation. Prognostic and predictive markers such as KRAS AND MSI can be used as the starting point for developing effective therapies. These new therapies can then be tested in clinical trials, even though only a few patients may qualify to participate.

Also, the combination of chemotherapeutic agents and monoclonal antibodies should be tested for their prospect in treating colon cancer. An example of such a combination was tested in a phase 3 randomized clinical trial by Tabernero et al., 2021 [177]. The study, with the clinical trial number NCT04737187, was tagged “SUNLIGHT”. It is a follow-up/confirmatory test on a previous study that reported the efficacy of the combination therapy, Trifluridine/tipiracil (FTD/TPI), and the monoclonal antibody, Bevacizumab (Bev) carried out by Prager et al. (2023) [178]. Study participants were randomized into two groups, with the first group receiving FTD/TPI+BEV while the other group received FTD/TPI. The primary outcome was overall survival, and the result showed that the combination of FTD/TPI and BEV produced an improvement in the overall survival, and the result was statistically significant when compared to the group that took FTD/TPI alone.

To compare the activity of FTD/TPI+BEV with the already approved fluoropyrimidine-irinotecan+ BEV, Kuboki et al. (2023) [179] conducted a randomized noninferiority trial. The experimental group received FTD/TPI+BEV, while the control group received fluoropyrimidine-irinoteacn+ BEV. The result showed that even though FTD/TPI+BEV did not demonstrate noninferiority over the control, the patients’ quality of life improved compared to the control. The result also showed that the combination could be a substitute for irinotecan-containing combination drugs, especially in patients who cannot tolerate irinotecan due to associated nonhematologic adverse effects. Thus, there is a prospect for improved efficacy and overall survival in the use of chemotherapeutic agents and monoclonal antibodies and should, therefore, be explored more.

Inhibitors of the vascular endothelial growth factor receptors (VEGFR), such as fruquitinib, are also being investigated, especially for colorectal cancers that have proven resistant to available chemotherapy. Dasari et al., 2023 [180], conducted a multi-centered randomized, double-blind phase 3 clinical trial tagged FRESCO 2 to examine the effectiveness of fruquitinib, which is a very potent and highly selective VEGFR inhibitor. The result showed that fruquitinib produced a significant overall survival in the patients when compared to the placebo, indicating that fruquitinib can be considered a treatment option for patients with refractory colorectal cancer. The analysis of the quality-of-life data is currently ongoing, and it is expected to further reinforce the clinical advantage of fruquitinib.

## 7. Impact of Circadian Rhythm on CRD Drug Delivery and Efficacy

Circadian clocks (CC), arising from the suprachiasmatic nucleus (SCN) and peripheral clocks, have been recognized in biological organisms for over three decades. CC serves as an adaptive method for organisms to respond to changes in their natural environment. Day and night alternations culminating in environmental light or other stimuli follow the laws of nature and induce the SCN to recognize the environmental changes and release signals to specific pathways for the activation of subordinate clocks that would produce corresponding adjustments to the 24 h period cycle (Shafi et al., 2019; Kinouchi and Sassone-Corsi, 2020) [181,182]. Recent studies suggest the potential regulatory effects of circadian clocks in various physiological processes that impact human health (Zhou et al., 2021) [183], including CRC.

An earlier study involving meta-analysis of individual data from three international phase III trials compared the outcome of chronomodulated therapies of 5-fluorouracil, leucovorin, and oxaliplatin administered in chronomodulated (chronoFLO) against conventional (CONV) infusions. A hazard ratio of 1.59 (1.30 to 1.75) for overall survival (*p* = 0.002) indicated that males lived significantly longer on chronomodulated chemotherapy than on conventional chemotherapy. Also, the sex versus schedule interaction effect provided a strong predictive factor for optimal treatment schedules. On the contrary, a systematic review and meta-analysis of data from six randomized controlled trials involving 1347 patients on chronomodulated chemotherapies generally did not show any advantages over conventional therapies with regard to overall survival, objective response rate, or toxicity (Brenner et al., 2014; Huang et al., 2016) [4,184]. Other recent studies have confirmed these observations but found a reduction in hematologic toxicity in the chronomodulated therapy group (Nassar et al., 2023) [185]. Kinouchi and Sassone-Corsi (2020) [182] concluded that an information gap still exists regarding the existence of a common core clock connection between cancer and pluripotent stem cells that may elucidate the impact of the circadian clock on critical stages of cellular differentiation. Despite the promising potential of chronomodulated administration of chemotherapeutics agents for optimizing treatment outcomes, this modality has not been widely implemented in many clinical situations.

## 8. Conclusions

As the scourge of colorectal cancer continues to ravage the human population, greater attention is being paid to its management, particularly due to recent changes in patient demography: CRC affects more people between the ages of 40 and 50, the productive capacity of most economies in many countries. The disease burden is more pronounced in underdeveloped countries with a lack of access to up-to-date equipment for early detection and diagnosis and inadequate clinical facilities for patient treatment. Although it is regarded as a lifestyle disease whose incidence is significantly impacted by a Westernized diet and sedentary way of life, a recent analysis of its social demographic context has shown more impact of its burden on low and medium SDI countries. While several treatment options exist for colorectal cancer, surgery and chemotherapy remain the mainstay of treatment despite the intractable disease remission and toxicity associated with these treatment strategies. The widespread availability of novel technologies for its monitoring, early diagnosis, and care will continue to support the fight against CRC. For example, elucidation of the genetic basis initiation, development, and progression of CRC has led to the introduction of biotechnology products that have had a significant impact on its treatment. Inhibitors of the vascular endothelial growth factor receptors (VEGFR) such as fruquitinib, trifluridine/tipiracil (FTD/TPI), and the monoclonal antibody (bevacizumab) have proven efficacious in the treatment of CRC cells that were previously resistant to available chemotherapies. Current understanding of the molecular basis of CRC and identification of specific cell surface receptors for peptide ligands will support the nanoparticle-based targeting of combination chemotherapeutic agents and/or monoclonal antibodies to the cells in ways that will minimize the off-target binding, enable high target specificity, and preserve normal cells. Preservation of normal cells will, in turn, lead to the abolition or reduction of adverse drug events, promote patient compliance, and produce an overall improvement in treatment outcomes.

## Figures and Tables

**Figure 1 pharmaceutics-15-02620-f001:**
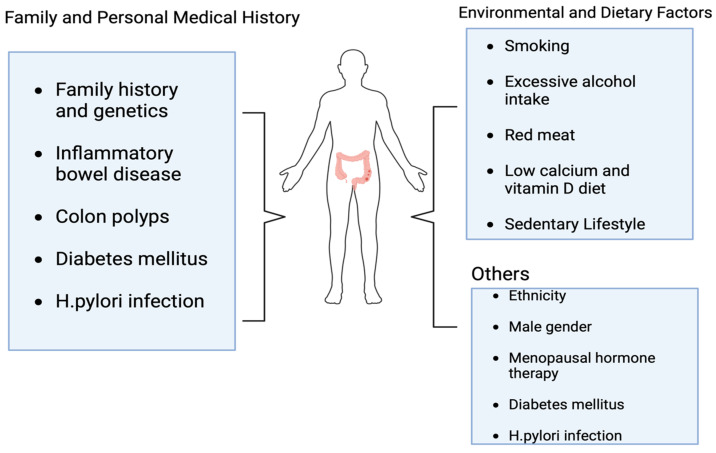
Risk factors of CRC. Created with BioRender.com.

**Figure 2 pharmaceutics-15-02620-f002:**
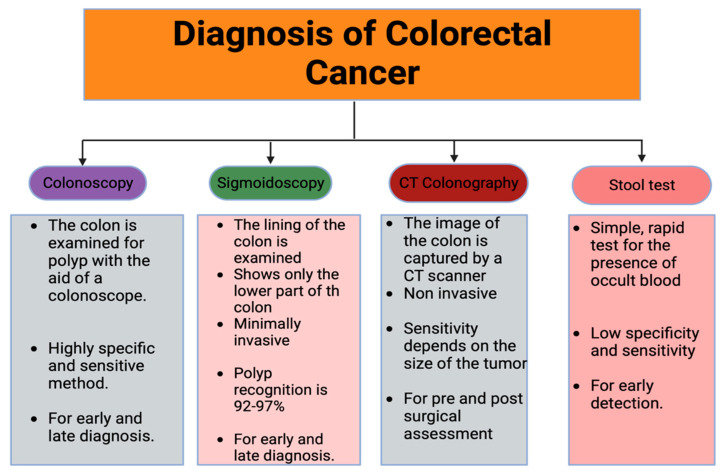
Diagnosis of CRC. Created with BioRender.com.

**Figure 3 pharmaceutics-15-02620-f003:**
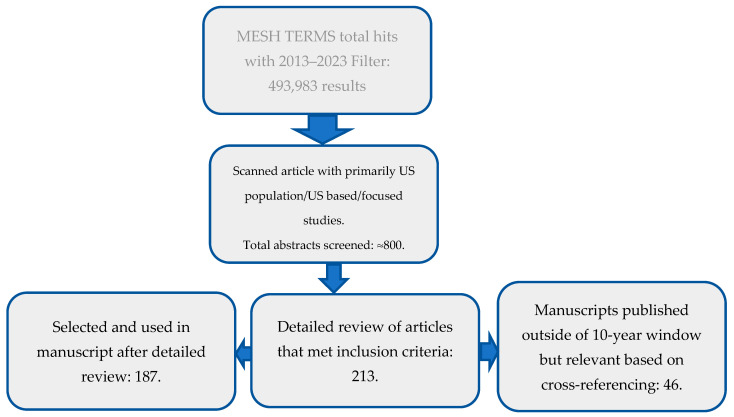
Reviewed articles selection protocol.

**Figure 4 pharmaceutics-15-02620-f004:**
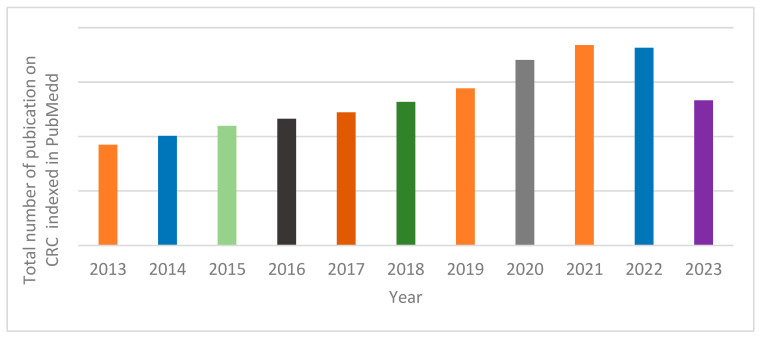
Research articles returned by MESH words used in the primary search in PubMed.

**Figure 5 pharmaceutics-15-02620-f005:**
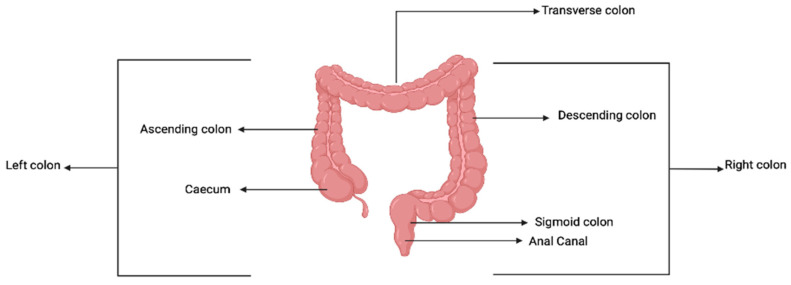
Anatomical description of the colon. Created with BioRender.com.

**Figure 6 pharmaceutics-15-02620-f006:**
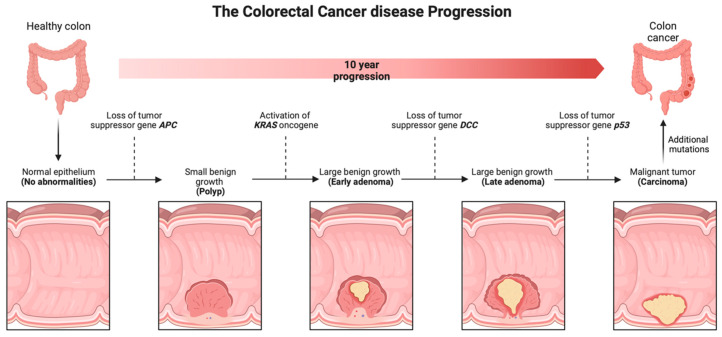
Disease progression of colorectal cancer. Created using a template on BioRender.com.

**Figure 7 pharmaceutics-15-02620-f007:**
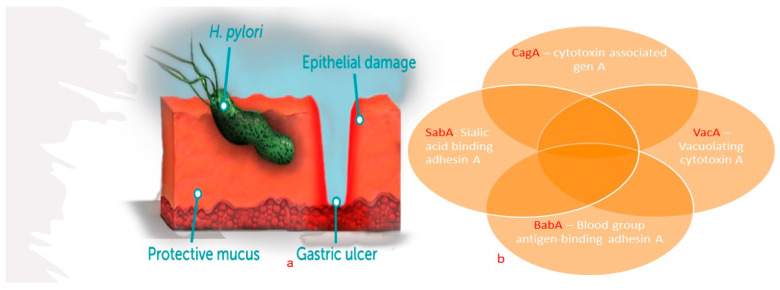
(**a**) Illustrative location of *H. pylori* in gastric mucosal. (**b**) Virulence factors that heighten cancer development (Chmiela et al. [55]).

**Figure 9 pharmaceutics-15-02620-f009:**
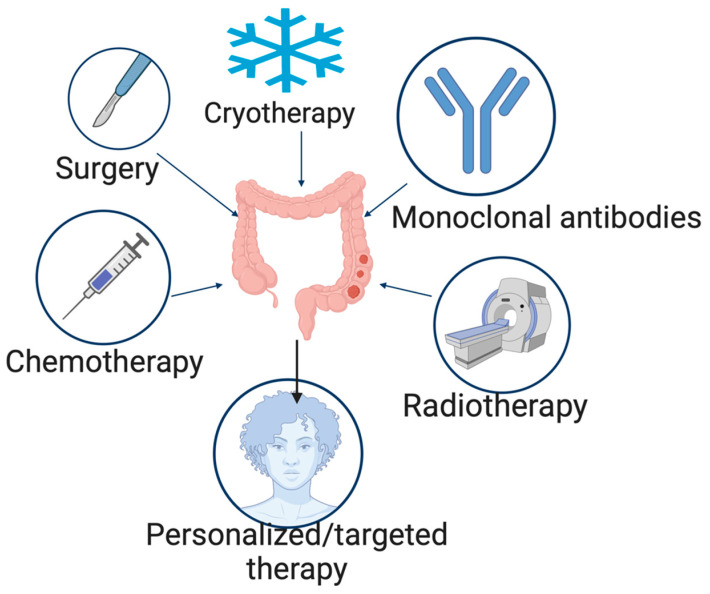
Treatment strategies for CRC. Created with BioRender.com.

**Figure 10 pharmaceutics-15-02620-f010:**
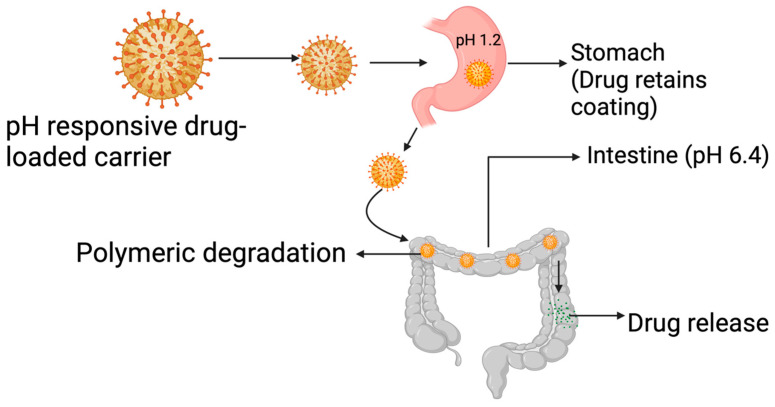
The pH approach to CRC drug delivery. Created with BioRender.com.

**Figure 11 pharmaceutics-15-02620-f011:**
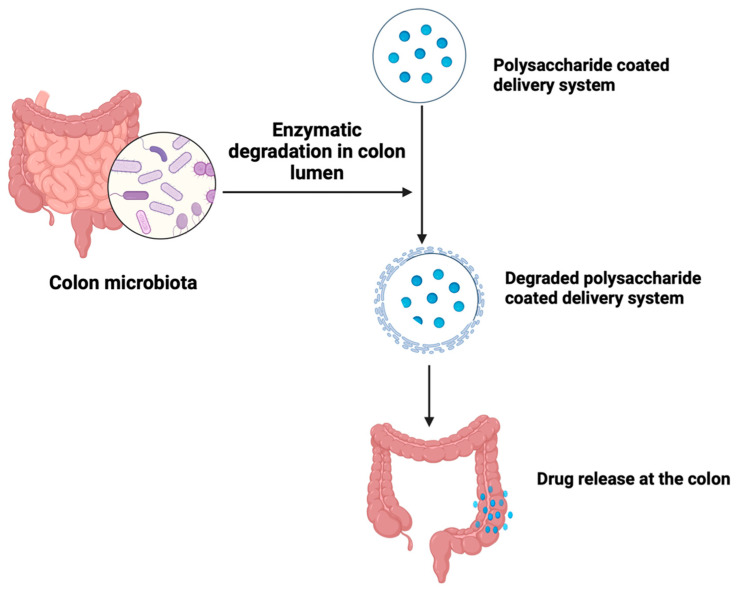
A description of the microbiota-based drug delivery. Created with BioRender.com.

**Figure 12 pharmaceutics-15-02620-f012:**
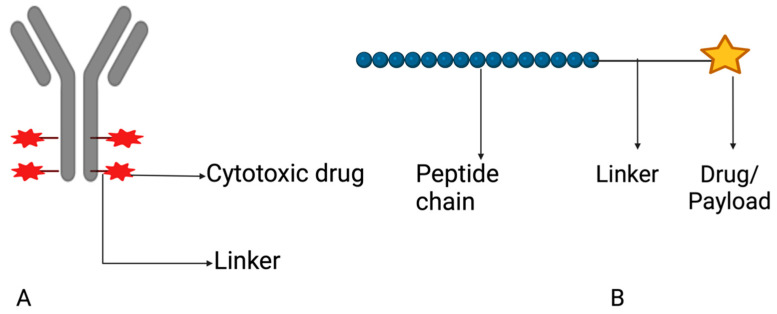
A description of an antibody–drug conjugate (**A**) and peptide–drug conjugate (**B**). Created with BioRender.com.

**Figure 13 pharmaceutics-15-02620-f013:**
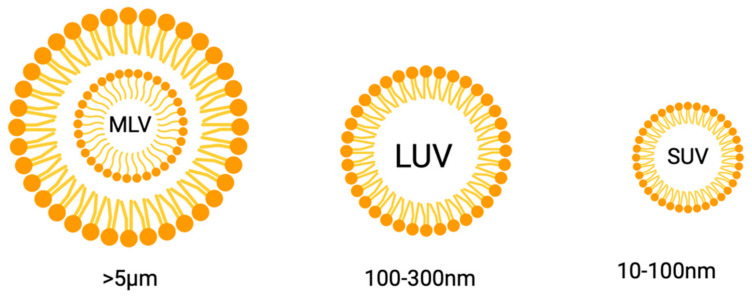
Various possible sizes of niosomes. Created with BioRender.com.

**Figure 14 pharmaceutics-15-02620-f014:**
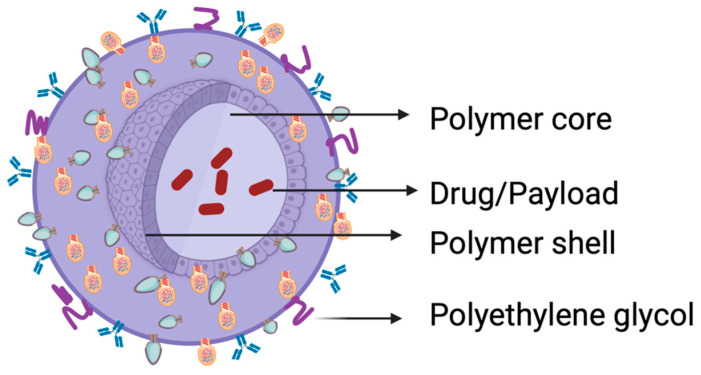
A polymeric nanoparticle for drug delivery. Created with BioRender.com.

**Table 3 pharmaceutics-15-02620-t003:** Drug-loaded vesicular system currently in clinical trial.

Vesicular System	Payload	Phase of Study	Clinical Trial Identifier	Sponsor
Liposome	Irinotecan and Bevacizumab	I	NCT05854498	University of Wisconsin (Madison, WI, USA)
	Irinotecan-based FOLFIRI combined with Bevacizumab	II	NCT05969899	Fudan University (Shanghai, China)
	Fluorouracil and Irinotecan and Leucovorin	I, II	NCT03337087	Academic and Community Cancer Research United (Rochester, MN, USA)
Polymeric nanoparticle	Cetuximab	I	NCT03774680	Al-Azhar University (Cairo, Egypt)

## Data Availability

Not applicable.

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
