# Peer review of "Colorectal Cancer: Disease Process, Current Treatment Options, and Future Perspectives"

_pharmaceutics, 2023, doi:10.3390/pharmaceutics15112620_

Round 1

Reviewer 1 Report

Comments and Suggestions for Authors

The introduction section appears to be quite brief and would benefit from a comprehensive revision to help readers clearly understand the scientific problems addressed by this research. To offer a comprehensive exposition of polymeric nanomaterials multifaceted attributes and applications, it is prudent for the authors to reference recent scholarly works authored by Ramiro Manuel Velasco Delgadillo, Mohan G. Kalaskar, EV Barrera, R.J. Linhardt, and Javier Villela Castrejón. These scholarly citations provide an invaluable foundation for probing the distinctive characteristics and merits of diverse nanoparticle variants and other biomaterials. 

Author Response

We thank the reviewer for their kind review of our manuscript and for the time and effort invested in the review. While we agree that significant advances are being made in the engineering of composite nanomaterials and carbon nanotubes, we did not see immediate relevance to delivery or targeting of anticancer drugs to colorectal cancer cells. It is therefore futuristic and we agree that using these carbonaceous biomaterials, which hitherto have been focused on tissue regeneration and engineering of biological sensors, will not only support therapeutic drug delivery and targeting but will help minimize adversed drug events by reducing off-target binding.

Having said that, we have incorporated recent work by Mamidi et al (2020, 2021a,b) in the 5.1.4.3. Polymeric nanoparticles, specifically in lines 741 to 746.

Thank you,

Dr. Adebayo

Reviewer 2 Report

Comments and Suggestions for Authors

Review provides insight into the current body of knowledge on etiology, progression, treatment (surgery, chemotherapy, radiation therapy, and cryotherapy) and overview future perspectives to combat this disease.

Though review is informative and well-written, it may benefit from clarifying some major statements and adding more novel strategies, as suggested below:

The first statement in the Introduction, “Globally, colorectal cancer (CRC) is the second most diagnosed cancer…” needs citation and should be checked as according to the other recent reviews it is the third most commonly diagnosed, and the second as a cause of cancer mortality (doi: 10.1038/s41575-019-0189-8; doi: 10.1016/j.tranon.2021.101174).

The authors are invited to extend statistical information from US to global trends.

  1. I suggest add a discussion on “birth cohort effects” and evidence of early-life exposure to colorectal cancer risk factors, particularly during fetal development, childhood, adolescence and young adulthood, a topic that have been accentuated in a recent review in Nature Rev. Gastroenterol Hepatol (doi: 10.1038/s41575-023-00841-9).

  2. While discussing putative genetic factors, the role of a well-established gene, APC (Adenomatous Polyposis Coli) should be included (e.g., doi: 10.1038/s41575-019-0189-8; doi: 10.1038/onc.2015.522; doi: 10.1093/jnci/djw332).

  3. The author’s statement regarding associations between colorectal cancer and country’s development, such stated in the Conclusion, P.20, L.677: “The disease burden is more pronounced in underdeveloped countries with a lack of access to up-to-date equipment for early detection and diagnosis” requires reconsideration or at least proper citation, since there are numerous reviews that came to an opposite conclusion (e.g., doi: 10.7759/cureus.33424; doi: 10.1016/j.tranon.2021.101174), and an another review, published in Nature Review Gastroenterol Hepatol in 2019 links colorectal caner risks with “westernezation” (doi: 10.1038/s41575-019-0189-8).

  4. This review may benefit from adding scope on novel approaches, incorporating knowledge of the circadian clock (doi: 10.1016/j.gendis.2022.05.014; doi: 10.3389/fphar.2021.741295) to therapeutic strategies to reduce toxicity / enhance efficacy of the treatment, known since 90s of the former century (doi: 10.1016/s0140-6736(97)03358-8). Treatment administered at hours of greater cell proliferation, i.e. morning hours may increase efficacy of early colon cancer (doi: 10.1177/11769351211067697). Chronomodulated chemotherapy can be considered to reduce haematological toxicities in advanced colorectal cancer (doi: 10.7759/cureus.36522). Efficiency of chemotherapeutic strategy on survival of patients with metastatic colorectal cancer can be higher in male patients (doi: 10.1093/annonc/mds148).

  5. Recently, ferroptosis-based therapeutic strategies achieving an increasing interest as a potential novel approach for treatment of cancers. This includes treatment of colorectal cancer as well (doi: 10.1038/s41416-023-02149-6; doi: 10.3389/fonc.2023.1059520; doi: 10.4251/wjgo.v15.i2.225; doi: 10.3390/biom13050820).

Author Response

The first statement in the Introduction, “Globally, colorectal cancer (CRC) is the second most diagnosed cancer…” needs citation and should be checked as according to the other recent reviews it is the third most commonly diagnosed, and the second as a cause of cancer mortality (doi: 10.1038/s41575-019-0189-8; doi: 10.1016/j.tranon.2021.101174).

  •  This claim has been revised in the introductory section as appropriate. Please see lines 8 and 9.

The authors are invited to extend statistical information from US to global trends.

  • I suggest add a discussion on “birth cohort effects” and evidence of early-life exposure to colorectal cancer risk factors, particularly during fetal development, childhood, adolescence and young adulthood, a topic that have been accentuated in a recent review in Nature Rev. Gastroenterol Hepatol (doi: 10.1038/s41575-023-00841-9).

  • This is outside the scope of the current discourse. We hope to submit a review on the CRC age-related risk factors in another manuscript under preparation.
  • While discussing putative genetic factors, the role of a well-established gene, APC (Adenomatous Polyposis Coli) should be included (e.g., doi: 10.1038/s41575-019-0189-8; doi: 10.1038/onc.2015.522; doi: 10.1093/jnci/djw332).

  • Addressed. Please see. line 64 - 76 in the revised manuscript attached.
  • The author’s statement regarding associations between colorectal cancer and country’s development, such stated in the Conclusion, P.20, L.677: “The disease burden is more pronounced in underdeveloped countries with a lack of access to up-to-date equipment for early detection and diagnosis” requires reconsideration or at least proper citation, since there are numerous reviews that came to an opposite conclusion (e.g., doi: 10.7759/cureus.33424; doi: 10.1016/j.tranon.2021.101174), and an another review, published in Nature Review Gastroenterol Hepatol in 2019 links colorectal caner risks with “westernezation” (doi: 10.1038/s41575-019-0189-8).

  • Addressed, please see section lines 40 to 52 in the revised manuscript. The subject is still very controversial but the contect of social demographic indices appears to be clarifying the distribution of CRC burden among regions.
  • This review may benefit from adding scope on novel approaches, incorporating knowledge of the circadian clock (doi: 10.1016/j.gendis.2022.05.014; doi: 10.3389/fphar.2021.741295) to therapeutic strategies to reduce toxicity / enhance efficacy of the treatment, known since 90s of the former century (doi: 10.1016/s0140-6736(97)03358-8). Treatment administered at hours of greater cell proliferation, i.e. morning hours may increase efficacy of early colon cancer (doi: 10.1177/11769351211067697). Chronomodulated chemotherapy can be considered to reduce haematological toxicities in advanced colorectal cancer (doi: 10.7759/cureus.36522). Efficiency of chemotherapeutic strategy on survival of patients with metastatic colorectal cancer can be higher in male patients (doi: 10.1093/annonc/mds148).

  • A paragraph has been added on potential impact of circadian rhythm on therapeutic outcome, please see line 800 to 828 in the attached document.
  • Recently, ferroptosis-based therapeutic strategies achieving an increasing interest as a potential novel approach for treatment of cancers. This includes treatment of colorectal cancer as well (doi: 10.1038/s41416-023-02149-6; doi: 10.3389/fonc.2023.1059520; doi: 10.4251/wjgo.v15.i2.225; doi: 10.3390/biom13050820).

  • Addressed. This is part of the targeting mcechanism with transferin as ligand. Please, see line 104 - 107 and lines 596 - 602 of the revised manuscript.

Reviewer 3 Report

Comments and Suggestions for Authors

This review aimed to provide insight into the disease progression, currently available treatment options, and their challenges, and provided some important insights into the future perspectives. This review paper is interesting and useful for scientists and physicians. I have the following comments:

1. In the abstract part the authors must write the used in your study the method of data collection, searching machines, keywords, and the period they collected data.

2. In the abstract part the authors must write the results from their data mining.

3. In the abstract part the authors must write the conclusion and recommendations for their study. In addition, these parts must be written in the manuscript.

4. The authors must follow the journal style regarding references.

5. To make your review more useful, I suggest adding a section regarding the side effects and complications of CRC treatments.

6. I suggest adding a section about the herbal materials used for the treatment and prophylaxis of CRC.

Author Response

  1. In the abstract part the authors must write the used in your study the method of data collection, searching machines, keywords, and the period they collected data.
    • Search method has been incorporated. A section on reviewe. methodology has been incorporated. please, see lines 130 -151.
  2. In the abstract part the authors must write the results from their data mining.
    • Done. Please see the revised abstract in the attached manuscript.
  3. In the abstract part the authors must write the conclusion and recommendations for their study. In addition, these parts must be written in the manuscript.
    • Done. Please see the revised abstract and conclusion in the attached revised manuscript.
  4. The authors must follow the journal style regarding references.
    • Referencing has been revised per journal style.
  5. To make your review more useful, I suggest adding a section regarding the side effects and complications of CRC treatments.
    • Done. Please see section 4.6. of the revised manuscript.
  1. I suggest adding a section about the herbal materials used for the treatment and prophylaxis of CRC.
    • To date, no regulatory approvaal has been given to an herbal medicine for the management of colorectal cancer. This therefore seems to be outside the scope of this review.

Reviewer 4 Report

Comments and Suggestions for Authors

The authors based their review on the premise that colorectal cancer (CRC) is one of the deadliest malignancies, ranking second only to lung cancer in men and third in women. Moreover, they claimed that CRC remains prevalent, and its incidence is expected to double by 2035. In addition, because CCR can be asymptomatic for several years, early detection and diagnosis are hampered. However, despite significant advances in the treatment of CRC, including surgery, chemotherapy, radiotherapy, and cryotherapy, several problems associated with treatment prevail, including unwanted effects, toxic side effects, and therapeutic failures. Overall, this study provides an overview of the progression of the disease, available treatment options, challenges, and some important insights into the future outlook.

This review has an impact on the field; however, a minor concern should be adequately addressed as the abstract requires revision in terms of the logical sequence of ideas. Please revise and amend it for a better understanding. In addition, thorough proofreading is recommended.

As the above may be corrected during the editorial process, this manuscript has been endorsed for publication in Pharmaceutics.

Comments on the Quality of English Language

Thorough proofreading is recommended.

Author Response

... abstract requires revision in terms of the logical sequence of ideas. Please revise and amend it for a better understanding. In addition, thorough proofreading is recommended.

  • We have revise the manuscript for organization, fluidity and logical sequencing. Please, see the attached revised manuscript.

Round 2

Reviewer 2 Report

Comments and Suggestions for Authors

The authors correctly addressed stated issues and improved review accordingly. They also added two important sections "Side effects and complications of CRC treatments" and "Impact of circadian rhythm on CRD drug delivery and efficacy". The review is recommended for publication.

Reviewer 3 Report

Comments and Suggestions for Authors

The authors conducted all the required corrections and I have no more comments